# Prevalence of Protein-Energy Wasting in Dialysis Patients Using a Practical Online Tool to Compare with Other Nutritional Scores: Results of the Nutrendial Study

**DOI:** 10.3390/nu14163375

**Published:** 2022-08-17

**Authors:** Marta Arias-Guillén, Silvia Collado, Elisabeth Coll, Jordi Carreras, Loreley Betancourt, Bárbara Romano, Marisol Fernández, Verónica Duarte, Julia Garro, Jordi Soler, Juan Carlos González, Jordi Calabia

**Affiliations:** 1Renal Transplantation and Nephrology Department, Hospital Clinic de Barcelona, 08036 Barcelona, Spain; 2Nephrology Department, Hospital del Mar-Parc de Salut Mar, 08003 Barcelona, Spain; 3Nephrology Department, Fundació Puigvert, 08025 Barcelona, Spain; 4Centre Diaverum Baix Llobregat, Diaverum, 08908 L’Hospitalet del Llobregat, Spain; 5Nephrology Department, Corporació Sanitaria Parc Taulí, 08208 Sabadell, Spain; 6Nutrition and Dietetic Unit, Hospital Clinic de Barcelona, 08036 Barcelona, Spain; 7Nephrology Department, Hospital de Terrassa, 08227 Terrassa, Spain; 8Nephrology Department, Hospital Joan XXIII, 43005 Tarragona, Spain; 9Nephrology Department, Hospital Germans Trias i Pujol, 08916 Badalona, Spain; 10Nephrology Department, Hospital de Mollet, 08100 Mollet del Vallés, Spain; 11Nephrology Department, Hospital Universitari de Girona Doctor Josep Trueta, 17001 Girona, Spain

**Keywords:** protein-energy wasting, online tool, dialysis, nutritional assessment, malnutrition inflammation score, subjective global assessment

## Abstract

This cross-sectional study aims to explore the prevalence of protein-energy wasting (PEW) in dialysis patients in Catalonia, Spain, using a new and practical online tool which enables rapid calculation and comparison with other nutritional scores. Methods: A web tool (Nutrendial) was created to introduce different variables and automatically calculate PEW, Malnutrition inflammation Score (MIS) and Subjective Global Assessment (SGA) in 1389 patients (88% in haemodialysis (HD)), 12% in peritoneal dialysis (PD) from different regions of Catalonia. Results: A prevalence of 23.3% (26% HD, 10.2% PD) of PEW was found, with a mean MIS score of 6 and SGA score of C in 7% of the patients. ROC analysis showed MIS as the best nutritional score to diagnose PEW (AUC 0.85). Albumin delivered lower diagnostic precision (AUC 0.77) and sensitivity (66%). A cut off point of 7 (86% sensitivity and 75% specificity) for MIS and 3.7 mg/dL for albumin were found to predict the appearance of PEW in this population. SGA B or C showed an 87% sensitivity and 55% specificity to diagnose PEW. Very low nutritional intervention (14%) was recorded with this tool in patients with PEW. Conclusions: This new online tool facilitated the calculation of PEW, enabling different professionals—including nephrologists, dieticians and nurses—to efficiently obtain insights into the nutritional status of the Catalonian dialysis population and implement the required nutritional interventions. MIS is the score with more sensitivity to diagnose PEW.

## 1. Introduction

Protein-energy wasting (PEW) [1] is the term proposed by the International Society of Renal Nutrition and Metabolism (ISRNM) to characterize the metabolic alterations related to uremia, malnutrition and hypercatabolism in patients with chronic kidney disease (CKD). PEW is usually associated with a lower functional capacity to adapt to stressful situations, worse morbidity, an increase in hospital admissions, a greater risk of infections and mortality [2]. In addition to PEW, there are other specific scores, such as the subjective global assessment (SGA) or malnutrition-inflammation score (MIS), proposed by Kalantar-Zadeh [3,4] that show prognostic value in patients on hemodialysis.

PEW is much more prevalent in dialysis than in pre-dialysis phases [5] because the dialytic procedure induces a net protein catabolic state, influenced by the dialysis technique and a systemic inflammatory response in relation to the biocompatibility of the system. Nutritional status is one of the main treatable factors affecting the prognosis and evolution of patients with CKD. Therefore, as established in the K-DOQI guidelines [6], the European Consensus [7], the Guidelines of the American Dietetic Society for the nutritional care of renal patients [8] and a recent review of the existing recommendations from the main clinical practice guidelines [9], the assessment of the nutritional status of patients with CKD in dialysis should be included in routine monitoring as it can be corrected, improves survival and reduces comorbidity and hospital admissions. However, the lack of dieticians in most of the renal care units [10,11] leaves nephrologists holding the main responsibility for the nutritional care of HD patients. This creates a need for a fast, practical and reliable method for assessing nutritional status, as this is one of the main barriers for the effective provision of this type of care.

In Catalonia, a registry to evaluate the malnutrition–inflammation status of our patients in chronic renal replacement therapy does not exist, so the exact prevalence of malnutrition is unknown, and nutrition intervention methods differ between centers. The objective of this study is to evaluate the prevalence of malnutrition in CKD patients in dialysis in Catalonia, using a new practical online tool to easily calculate the prevalence of PEW and compare it to different scores (MIS and SGA).

## 2. Materials and Methods

### 2.1. Study Design and Subjects

We designed a cross-sectional, observational and descriptive multi-center study, including patients from different geographical areas of Catalonia in renal replacement therapy: both hemodialysis (HD) and peritoneal dialysis (PD). This included all men and women over 18 years of age who started renal replacement therapy at least 12 weeks before the study began. The project was approved by the Ethics Committee of all participating centers, in accordance with the Declaration of Helsinki, as well as the Organic Law on Data Protection (15/1999), which aims to guarantee and protect public freedoms and fundamental rights of natural persons, and especially the personal and family privacy of all patients who participated in the study.

Once the potential patients for this study were identified, their written consent was sought after having provided them with detailed information on the study characteristics; their duties, rights and the potential benefit that research would bring. Through this process, 1223 patients in HD and 166 in PD were recruited for the study.

Different demographic and anthropometric variables, clinical history related to renal replacement therapy, functional capacity, dietary intake (according to MIS/SGA criteria) and gastrointestinal symptoms were collected at the time of inclusion. Major comorbidities (CCM) were identified as congestive heart failure III-IV, severe ischemic heart disease, AIDS, moderate-severe chronic obstructive pulmonary disease, major neurological sequelae, neoplasia with metastasis or recent chemotherapy. The patient was then classified as no comorbidity, without CCM, moderate CCM if one CCM was present and severe CCM if two or more were present. Fat and muscle reserves were assessed by physical examination and analytical variables with the aim of applying the diagnosis of PEW (currently defined as the gold-standard) and for calculating the MIS and SGA scores. To predict PEW, the Nutrendial web application applies an algorithm including the variables needed to calculate PEW according to the classical criteria [12] for each body nutrition compartment, specifically.

For the biochemical parameters section, albumin < 3.8 g/dL or cholesterol <100 mg/dL were used.For the body mass section, BMI < 23 kg/m^2^ or a weight loss > 10% in the last 6 months were used.For the muscle mass section, creatinine < 8 mg/dL was used [13].For the dietary intake section, nPCR < 0.8 g/kg/day was used.

The data were collected during the second HD session of the week or in the peritoneal dialysis control. A form was provided to facilitate data collection (Figure 1). In the physical examination section, visual support (photographs) was available in each answer option to aid in decision-making.

### 2.2. Data Collection

A web application (www.nutrendial.cat) was created for the introduction of the different variables that were stored in a centralized database (Figure 2). The different participating units were responsible for maintaining confidentiality. The treatment, communication and personal data transfer of all participating subjects complied with the following Spanish organic laws: 15/1999 of 13 December on personal data protection; and 14/2007 of 3 July on biomedical research. The recruited patients were coded alphanumerically to guarantee anonymity. Their personal identity was not to be disclosed, except in cases of medical emergency or legal requirement. Patients included in this study were recruited between January 2018 and December 2019 from different hospitals and hemodialysis centers in Catalonia (Appendix A).

### 2.3. Statistical Analysis

The web application allows data export for the subsequent statistical analysis, using the software statistical package (SPSS, version 22.0, IBM corp, NY, United States). Prevalence rates of malnutrition were determined according to the different scores. Descriptive statistics were carried out, defining the continuous variables by mean and standard deviation. A comparison of continuous variables was performed by using the Student’s *t*-test for unpaired data, and the Chi-square test for qualitative variables as well as ANOVA, depending on the number of groups. If continuous variables did not show a normal distribution, the data were analyzed using the Mann–Whitney’s U test. A *p*-value < 0.05 was considered significant.

## 3. Results

### 3.1. Clinical Characteristics According to PEW Diagnosis

A total of 1389 dialysis-dependent patients were enrolled in this study. The median age was 72 years (IR 60–81), 63.4% of patients were men and 36.1% had diabetes mellitus. A total of 88% of participants were on hemodialysis (76% online hemodiafiltration) and 12% on peritoneal dialysis (50% automated PD). The median of time on dialysis was 31 months (RI 13–66). Regarding nutrition parameters, the mean of BMI, albumin, TIBC and cholesterol was 25.72 ± 5.15 kg/m^2^, 3.70 ± 0.41 g/dL, 237.63 ± 52.06 mg/dL and 151.98 ± 37.77 mg/dL, respectively.

The prevalence of PEW of the whole population was 23.3%, with a median of the MIS score at 6 (RI 4–9) and 7% of patients with an SGA score of C. The baseline characteristics of the study population according to the PEW diagnosis are presented in Table 1. Patients catalogued with PEW were older than those without PEW (73 ± 14 vs. 68 ± 15 years, *p* < 0.001), but there were no differences in gender or time on dialysis. As expected, all the serum nutritional parameters analyzed were significantly lower in the PEW population, including the protein catabolic rate (nPNA). Compared to those on hemodialysis, patients on peritoneal dialysis had a lower percentage of PEW (10.2% vs. 25.3%, respectively, *p* < 0.001) and were also significantly younger, with less time on dialysis, greater body mass and with more comorbidities, but with better functional capacity (*p* < 0.05) (Table 1).

Patients with PEW presented at least one CCM (40.2%) and had two or more CCMs (8.4%) compared to patients not classified as PEW, of which 61.5% had no moderate or major comorbidity registered, while 28.5% had one CCM (*p* < 0.001). Related to functional capacity, 25.1% of the PEW patients were classified in the Nutrendial tool as “normal” in comparison with 59% of those without PEW. Patients with PEW presented significantly greater fatigue, experienced some difficulty walking and almost 17% reported not doing any type of physical activity (*p* < 0.001).

With respect to nutritional parameters, worse food intake was observed in the PEW population, especially related with suboptimal solid food intake (52.9% vs. 17.1%). Moreover, gastrointestinal symptoms, such as occasional nausea, vomiting, diarrhea or anorexia were present in 61.5% of the patients diagnosed with PEW vs. 22.3% in patients without PEW. Physical examination showed that subcutaneous fat loss (moderate-severe in 47.6% vs. 9.1%) and signs of muscle loss (moderate-severe 48.3% vs. 9.5%) were higher in the PEW group (Table 2).

Nutritional intervention was also recorded in the Nutrendial online tool, and the results showed that patients with PEW received a higher volume of interventions (14% vs. 9.8%) (Table 1). More specifically, 53% patients with PEW were receiving oral nutritional supplements (ONS) and 8% were receiving intradialysis parenteral nutrition (IDPN), while 22% of the patients not catalogued with PEW were receiving ONS and 1% were receiving IDPN.

### 3.2. SGA, MIS Scores and Other Nutritional Parameters as Predictors of PEW

Cross tabs were performed to evaluate the ability of the SGA score as a predictor of PEW, and the group classified as VGS C had a specificity to diagnose PEW of 100% and a sensitivity of 30%, but when we combined the groups classified as B/C, the sensitivity increased to 88%, although the specificity decreased to 55%.

ROC curve analysis was also performed to evaluate the ability of MIS score as a predictor of PEW (Figure 2). The calculated area under the curve was 0.85 (95% confidence interval: 0.82 to 0.87, *p* < 0.001). Analysis of the MIS score enabled the identification of a cut off value after which the appearance of PEW became more probable. For this Catalonian dialysis population, the calculated value is seven, with a sensitivity of 86% and specificity of 75.3%.

This ROC curve analysis was also performed with other nutritional parameters: albumin predicted malnutrition with a value of 0.77 (95% CI: 0.741–0.801) and a cut-off point for predicting the appearance of PEW in this population of 3.71 mg/dL, with lower sensitivity (66%). Body mass index (BMI) proved a more reliable indicator for the prediction of PEW with a value of 0.789 (95% CI: 0.760–0.818), greater sensitivity (88%) and a cut-off point for this population of 22 kg/m^2^ (Figure 3, Table 3).

## 4. Discussion

This is the first study analyzing the nutritional status in the Catalonian dialysis population, using a new web tool (www.nutrendial.cat) which offers an easy-to-manage method to calculate the prevalence of PEW (classical criteria ISRNM) as well as nutritional scores, such as SGA and MIS, also allowing comparison between them. We observed that 23.3% of the patients on dialysis included in our study presented PEW, obtaining the best PEW diagnostic precision with MIS, with the albumin being far apart.

PEW prevalence in kidney disease patients was, until very recently, poorly and inconsistently evaluated, with wide ranges such as 18–75% reported, probably due to a lack of standardized PEW definitions, a variety of existing assessment tools and cut-off points, small sample size studies and differences in the socioeconomic realities of the countries in which the studies took place [14].

A recent meta-analysis of 90 studies covering 16,434 dialysis patients from 10 geographical regions, using only SGA or MIS scoring systems, found large variations in PEW prevalence results: in HD, scores ranged between 28–56% (median 43%) and in PD patients, between 32–49% (median, 36%) [14]. In our population, lower prevalence of PEW in HD (25.3%) and in PD (10.2%) was observed, appearing closer to the results obtained by Gracia-Iguacel and colleagues in a Spanish cohort from Madrid of 122 prevalent HD patients [15], where 37% at a baseline visit had PEW. Another Spanish cohort from the Canary Islands of 468 prevalent HD patients [16] showed that 23% had PEW, according to the ISRNM criteria.

Ashabi et al. were the first group to compare various scoring methods (SGA, dialysis malnutrition score (DMS) and MIS) for the diagnosis of PEW in 291 HD patients [17] showing that, based on SGA, 60.5% of Tehran HD patients had mild-to-moderate PEW and 1% had severe PEW. DMS and MIS were very similar to SGA in identifying PEW in HD, with a sensitivity and specificity of 94% vs. 87% and 88% vs. 96%, both with an AUC of 97%, and the same sensitivity and cut-off (seven) as was obtained for MIS as a predictor of PEW. In Spain, a study of 186 patients [18] showed that the prevalence of PEW was 30.1% using the classic criteria, while 27.9% of SGA values were within the range of malnutrition and no differences were found between the two methods.

Like the Catalonian sample, in the Madrid cohort, the group did not observe significant differences by gender, and they detected less prevalence of PEW in the diabetic population [19]. In the Canarian group, patients with PEW showed a longer dialysis vintage but, in our population, we did not observe significant differences with respect to vintage. As in other studies, elderly patients were much more vulnerable to PEW development [20,21]. Related to comorbidities, as expected, the patients catalogued as PEW in our population had a significantly greater number of major CCM.

Among the biochemical markers available for the diagnosis of PEW in patients with CKD, it is widely known that there is no nutritional marker that meets all the requirements as such. Low serum albumin is a strong predictor of mortality, but this does not necessarily equate to a diagnosis of malnutrition. The cut-off points suggested are often based on survival analysis, and its levels can be altered by causes other than nutritional deficit (volume overload, inflammation, comorbidities as well as changes in synthesis and degradation of this protein). On the other hand, given its wide body distribution and half-life (15–20 days), it responds slowly to alterations in visceral protein reserves, so it would be a late marker of malnutrition and, therefore, not very practical for early nutritional interventions. In addition, studies in the older population without CKD have shown that serum albumin levels tend to be lower than in the younger population [22].

All the serum nutritional parameters in our study were significantly lower in the PEW group, but we were interested in evaluating these different serum parameters, as well as the BMI and MIS as possible predictors of PEW. We also calculated the cut-off values for our Catalonian dialysis population, based on the considerations of other authors who conveyed that the cut-off points of most malnutrition parameters proposed by ISNMR to calculate PEW derive from American populations, and it may be that they cannot be extrapolated to other geographic areas and lifestyles, such as Europe, Asia or, in our case, Mediterranean countries [15,23]. In our study, the cut off value calculated for the albumin was 3.7 g/dL, obtaining a moderate sensitivity (66%) but a higher specificity (81%), and for the BMI was it was 22 kg/m^2^, obtaining very good sensitivity (88%) and a moderate specificity (81%); both very similar to the cut-off specified in the criteria for the clinical diagnosis of PEW (albumin < 3.8 g/dL in case the bromocresol green kit was used to measure and BMI < 23 kg/m^2^) [2]. In a study performed in the Renal Patient Register of Catalonia (RMRC) including 6290 dialysis patients, as many as 14.4% of patients were in the range of underweight (BMI < 20 kg/m^2^), while 14.3% were obese (BMI > 30 kg/m^2^) [24], confirming that in our Mediterranean population over > 65 years, overweight and obese patients, as defined by the WHO, had better survival. In our study 15.7% had BMI > 30 kg/m^2^, very similar to the data previously reported.

Only 14.2% of the patients with PEW received nutritional intervention. Nutritional support is shown to improve survival markers, protein intake or promote changes in body composition with increased lean mass [22]. Many of the patients who were candidates for a nutritional intervention probably did not receive it, either because it is challenging to correctly diagnose PEW [25] or because many patients are not routinely seen by a dietician [26].

PEW criteria are initially attractive due to their multifactorial scope (analytical, anthropometrical and nutritional), but implementation in clinical practice might be difficult and time consuming for health professionals, including dieticians. A recent study based on a survey related to kidney nutrition care, answered by key kidney care stakeholders in ISN-affiliated countries, revealed that only 48% of the 155 countries have dieticians/renal dieticians to provide this specialized service. Important deficiencies in interdisciplinary communication on kidney nutrition care delivery were also detected, especially in lower-income countries [26]. Integrated approaches by physicians and dieticians are needed to take a holistic view of a patient’s nutritional assessment and support beyond just the control of particular laboratory parameters [27].

Easy-to-use tools to identify patients at malnutrition risk are critically needed by the renal community. Some time ago, we suggested bioimpedance spectroscopy as a practical tool for early detection and prevention of PEW in hemodialysis patients [28], and with this online tool (Nutrendial), we wanted to go a step further to enable professionals who work with CKD patients to screen and monitor PEW following the ISRNM criteria. In comparison with other tools, Nutrendial is easy to use, decreases the margin of human error—as the calculations of PEW, MIS and SGA variables are automatic—and consequently, the time employed to calculate the nutritional status of CKD patients. Individual measurements remained for every patient to follow their evolution and generate a final report that can be given to the patient or other professionals. The online tool allows for the immediate exploitation of data and provides a commons source of information to enable the execution of multicenter studies. It is currently accessible only to Spanish health professionals but will be translated to English in order to offer it to the international community.

We did not compare the data with bioimpedance as it was not available for all dialysis centers at the start of the study.

This study demonstrates the strength of the Nutrendial online tool as a streamlined means of obtaining a final nutritional result. The study identifies a good correlation between other widely used scores to detect the PEW status. The study uses a sample size of 1389 dialysis patients, representative in age and etiology of CKD patients in dialysis as shown in Table 1. Furthermore, for the first time, we were able to establish cut-off points for the main nutritional parameters in our population to help different professionals to detect PEW in a consistent way. A possible limitation in the interpretation of the present study could be its observational nature that precludes the assessment of causality.

## 5. Conclusions

The assessment of nutritional status for the detection and management of PEW is strongly recommended in CKD patients and should be included within the usual practices of a dialysis unit. We have developed an effective online tool that enhances the ability of different healthcare professionals—nephrologists, nutritionists, nurses—to diagnose PEW, better enabling them to implement the required nutritional interventions which can improve quality of life and survival rates among dialysis patients.

## Figures and Tables

**Figure 1 nutrients-14-03375-f001:**
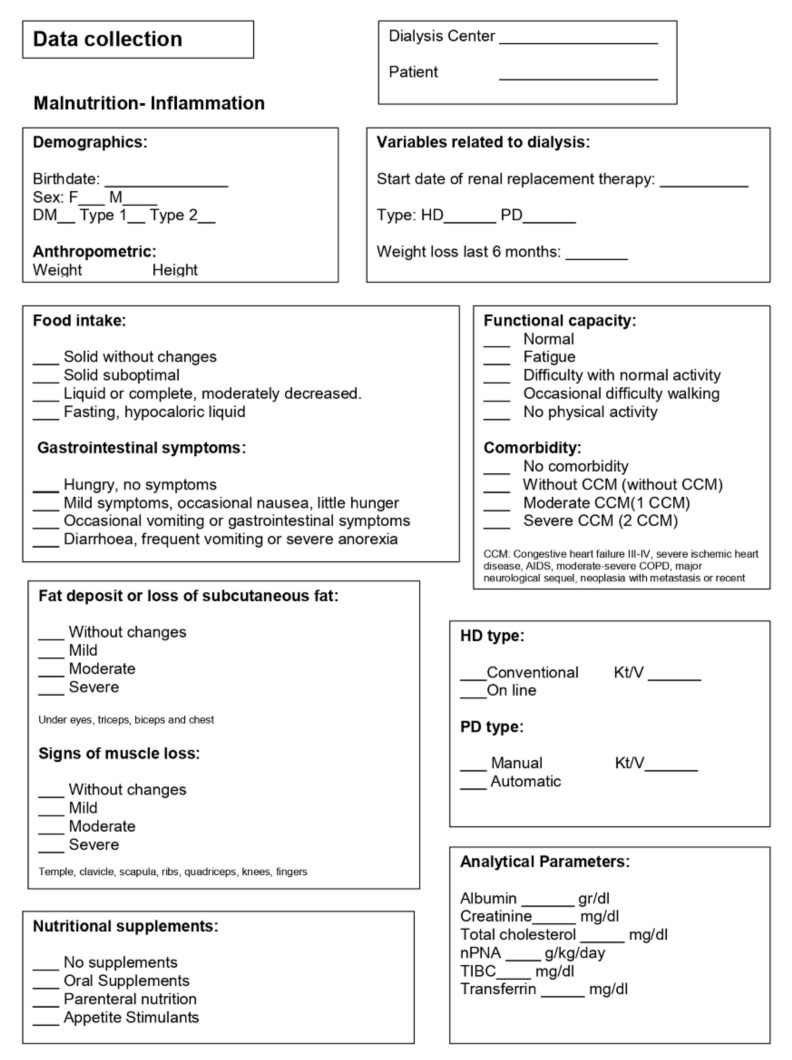
Data collection sheet.

**Figure 2 nutrients-14-03375-f002:**
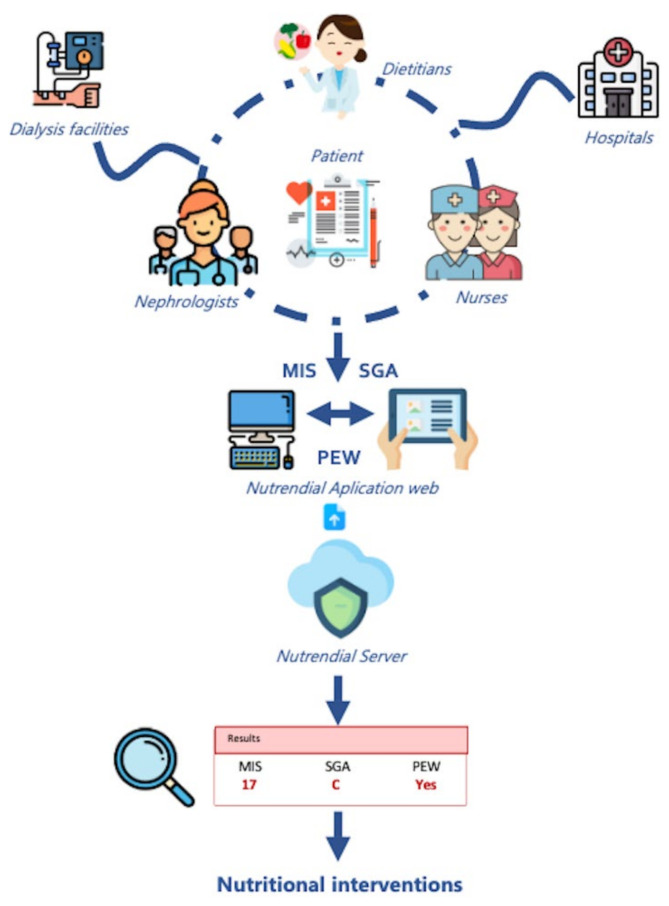
Nutrendial Web Application. Icons drawn by Surang, Freepikcompany, Paulalee, Flat icons, Ultimatearm and Eucalyp from www.flaticon.com (accessed on 29 March 2022). Figures shown in results box are examples only.

**Figure 3 nutrients-14-03375-f003:**
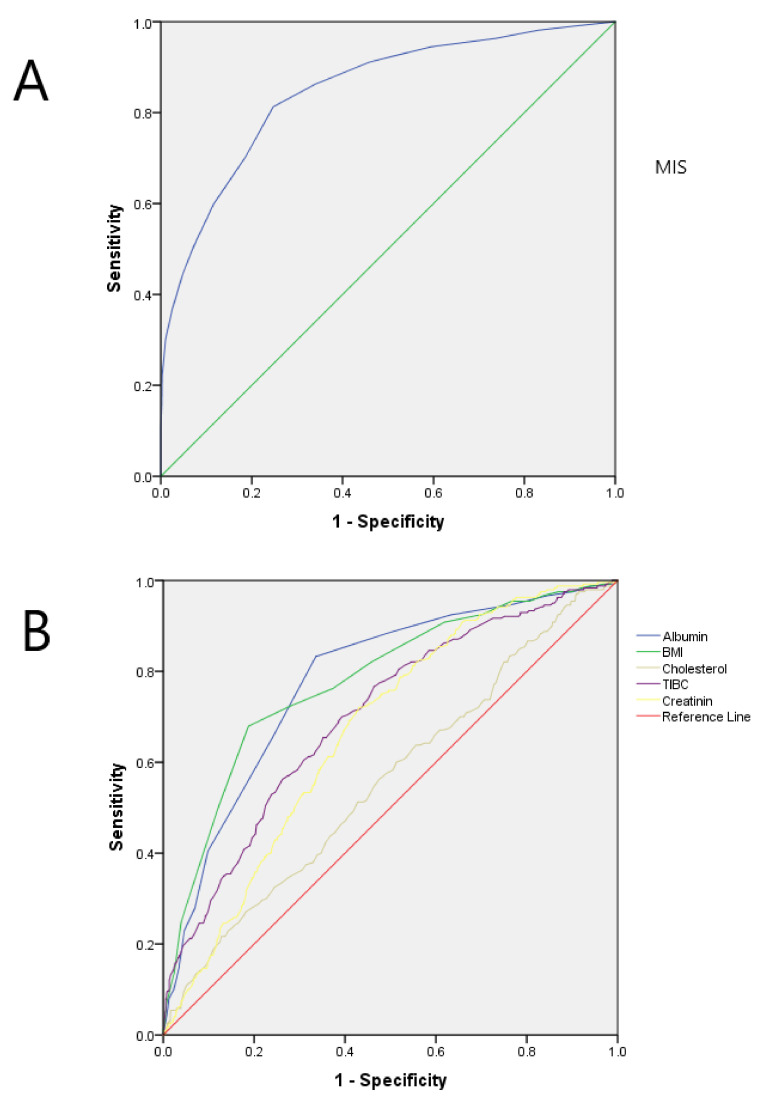
(**A**) Receiver operating characteristic (ROC) curve for malnutrition inflammation score (MIS) for the diagnosis of protein-energy wasting (PEW). Area under the curve (AUC) for MIS is 0.85 with *p* < 0.001 (**B**) Aggregated receiver operating characteristic (ROC) curves for several malnutrition markers for the diagnosis of protein-energy wasting (PEW). Area under the curve (AUC) for every marker are detailed on Table 3. BMI, body max index; TIBC, total iron binding capacity.

**Table 1 nutrients-14-03375-t001:** Population characteristics according to the presence of PEW.

Characteristic	Without PEW (*n* = 1066)	With PEW (*n* = 326)	*p* Value
Age, year	70 (58–80)	76 (65–83)	<0.001
Men, *n* (%)	688 (64.5)	192 (59.4)	0.096
Diabetes mellitus (%)	405 (38.1)	111 (34.1)	0.346
% HD/DP	74.7/89.8	25.3/10.2	<0.001
Time on dialysis (months)	31 (13–65)	31 (12–71)	0.958
kt/v on target	886 (88.4)	275 (91.4)	0.151
BMI, kg/m^2^	26.81 ± 4.97	22.08 ± 3.90	<0.001
Albumin, g/dL	3.83 ± 0.33	3.50 ± 0.37	<0.001
Total cholesterol, mg/dL	153.68 ± 37.98	145.92 ± 36.43	0.002
Creatinine, mg/dL	7.34 ± 2.39	6.11 ± 1.96	<0.001
TIBC, mg/dL	244.78 ± 51.06	212.59 ± 47.71	<0.001
nPCR	1.11 ± 0.33	1.01 ± 0.33	<0.001
No commorbidities (%)	232 (21.8)	25 (7.7)	<0.001
MIS ≥ 8, *n* (%)	266 (24.9)	263 (80.8)	<0.001
SGA C (%)	0 (0)	97 (100)	<0.001
Nutritional intervention (%)	105 (9.8)	46 (14.2)	0.26

Clinical characteristics are presented as mean ± SD for continuous variables and *n* (%) for categorical variables. Variables with non-normal distributions are shown as median (interquartile range). *p* values were derived from the independent *t*-test/Mann–Whitney U test for continuous variables and chi-square test for categorical variables. PEW, protein-energy wasting; BMI, body max index; TIBC, total iron binding capacity; nPCR, normalized protein catabolic rate; MIS, malnutrition inflammation score; SGA, subjective global assessment.

**Table 2 nutrients-14-03375-t002:** Clinic parameters of nutrition.

	No PEW	PEW	*p* Value
**Food intake**			<0.001
Solid without changes	881 (82.6%)	125 (38.7%)	
Solid suboptimal	182 (17.1%)	171 (52.9%)	
Liquid or complete, moderately decreased	3 (0.3%)	26 (8%)	
Fasting, hypocaloric liquid	0	1 (0.3%)	
**Gastrointestinal symptoms**			<0.001
Hungry, no symptoms	838 (78.6%)	124 (38.4%)	
Mild symptoms, occasional nausea, little hunger	216 (20.3%)	150 (46.4%)	
Occasional vomiting or GI symptoms	11 (1%)	40 (12.4%)	
Diarrhea, frequent vomiting or severe anorexia	1 (0.1%)	9 (2.7%)	
**Fat deposits or subcutaneous fat loss**			<0.001
Without changes	645 (60.5%)	56 (17.3%)	
Mild	323 (30.3%)	113 (35%)	
Moderate	92 (8.6%)	118 (36.5%)	
Severe	5 (0.5%)	36 (11.1%)	
**Signs of muscle loss**			<0.001
Without changes	574 (53.8%)	48 (14.98%)	
Mild	391 (36.7%)	119 (36.8%)	
Moderate	95 (8.9%)	127 (39.3%)	
Severe	6 (0.6%)	29 (9%)	

**Table 3 nutrients-14-03375-t003:** ROC curve analysis performed to evaluate prediction of PEW.

Parameter	Area under Curve (95% Confidence Interval)	Cut-Off Value	Sensibility	Specificity
MIS	0.85 (0.82–0.87)	7	86%	75%
BMI, kg/m^2^	0.79 (0.76–0.82)	22	88%	67%
Albumin, g/dL	0.77 (0.74–0.80)	3.7	66%	81%
Total cholesterol, mg/dL	0.56 (0.52–0.60)	147	57%	52%
TIBC, mg/dL	0.70 (0.66–0.73)	226	70%	61%
Creatinine, mg/dL	0.66 (0.63–0.70)	7.4	81%	45%

MIS, malnutrition inflammation score; ROC, receiver operator characteristic.

## Data Availability

Not applicable.

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
