# Peer review of "Prevalence of Protein-Energy Wasting in Dialysis Patients Using a Practical Online Tool to Compare with Other Nutritional Scores: Results of the Nutrendial Study"

_nutrients, 2022, doi:10.3390/nu14163375_

Round 1

Reviewer 1 Report

Comments to Author
The authors conduct a cross-sectional study to evaluate the prevalence of protein-energy wasting (PEW) in dialysis patients (n=1389) including hemodialysis and peritoneal dialysis in Catalonia using a new web diagnostic tool (or classical criteria ISRNM) to calculate PEW and comparing to malnutrition inflammation score (MIS) and subjective global assessment (SGA). This study provides clinical characteristics of PEW and evaluate the SGA, MIS scores and other nutritional parameters as predictors of PEW. The authors identified using
SGA classified as B / C or MIS cut off value of 7 will increase the sensitivity. However, several comments should be clarified to make it clear.

Major point:

1.      The diagnostic criteria or scores for PEW from this web tool (or classical criteria ISRNM) is not shown. How to predict the PEW using this new system?  

2.      What is the prevalence of PEW in dialysis patients using traditional criteria SGA or MIS? Could authors provide the similarities or differences between this new diagnostic tool and traditional criteria?

3.      In Table 1, they evaluate the number and percentage of MIS<8 and SGA C between without PEW and with PEW is not clear. Consider changing to MIS>8 representing the PEW percentage?

4.      The authors reported very low nutritional intervention (14%) was recorded with this tool in patients with PEW. However, the nutritional intervention for dialysis patients without PEW is still very low (9.8%). It will be better to show how to early intervention in these patients. In Figure 2, it seems to show that only patients with MIS>17, SGA class C needs nutritional intervention. Please clarify this issue.  

5.      In ROC curve analysis for MIS score and other nutritional parameters such as albumin, BMI…etc, but these traditional bio-serological makers have already known less sensitivity. Could the authors provide the ROC curse for new web diagnostic tool?

6.      Discussion section should focus on this new web tool and what is the advantage or disadvantage compared to traditional tools?

Minor point:

1.      In line 136, there is a “space” between catalogued and with. Please correct it.

2.      This web tool is not open to assess and needs to be registered. Is it only a registration system in Catalonia Society of Nephrology? Can it apply to other countries? 

Author Response

Response to the comments of reviewer 1.

 Thank you very much for your comments and suggestions which we have taken into consideration and we have now incorporated into the revised manuscript.

Major comments

  1. The diagnostic criteria or scores for PEW from this web tool (or classical criteria ISRNM) is not shown. How to predict the PEW using this new system?  

Response 1: To predict PEW, Nutrendial application web applies an algorithm where the variables needed to calculate PEW according to the classical criteria (12) for each body nutrition compartment have been previously introduced, specifically:

  • For the biochemical parameters section, albumin < 3.8 g/dL or cholesterol <100 mg/dL were used
  • For the body mass section, BMI <23 kg/m2 or a weight loss >10% in the last 6 months were used
  • For the muscle mass section, creatinine < 8mg/dL was used (13)
  • For the dietary intake section, nPCR <0.8 g/kg/day was used

This information has been included in the main text, in Methods and references have been updated.

  1. What is the prevalence of PEW in dialysis patients using traditional criteria SGA or MIS?

Response 2.1: It is well known that the prevalence of malnutrition differs according to the stage of kidney disease, the dialysis technique and the methodology used for its diagnosis; Ashabi et al were the first group to compare various scoring methods (SGA, Dialysis malnutrition score -DMS- and MIS) for the diagnosis of PEW in 291 HD patients (17), showing that, based on SGA, 60.5 % of Tehran HD patients had mild-to-moderate PEW and 1 % had severe PEW. DMS and MIS were very similar to SGA for identifying PEW in HD with a sensitivity and specificity of 94 % vs 87% and 88 vs 96%, both with an AUC of 97%, and the same and the same sensitivity and cut-off (seven) as us was obtained for MIS as predictor of PEW

In Spain, a study of 186 patients (5) showed that the prevalence of PEW was 30.1% using the classic criteria while 27.9% of SGA values were within the range of malnutrition and no differences were found between the 2 methods.

This information has been included in the main text, in Discussion and references have been updated.

 Could authors provide the similarities or differences between this new diagnostic tool and traditional criteria?

Response 2.2: All the MIS and SGA scoring criteria are included in the Nutrendial web application and it calculates them automatically, there are no differences between this new diagnostic tool and traditional criteria.

  1. In Table 1, they evaluate the number and percentage of MIS<8 and SGA C between without PEW and with PEW is not clear. Consider changing to MIS>8 representing the PEW percentage.

Response 3: As the reviewer suggests, the number and percentage of MIS<8 and SGA C without PEW and with PEW were not clear, so the MIS value has been changed in Table 1 to ≥8, representing the PEW percentage.

  1. The authors reported very low nutritional intervention (14%) was recorded with this tool in patients with PEW. However, the nutritional intervention for dialysis patients without PEW is still very low (9.8%). It will be better to show how to early intervention in these patients. In Figure 2, it seems to show that only patients with MIS>17, SGA class C needs nutritional intervention. Please clarify this issue.  

Response 4: The Nutrendial study was originally designed to assess the prevalence of PEW in CKD patients on dialysis in Catalonia using a new online tool to easily calculate and compare it with different scores (MIS and SGA). It was not intended to be an interventional study, so we only collected the nutritional intervention provided by the different dialysis centers, that now is more detailed in the text:

The following sentence has been included in the results section: More specifically, 53% patients with PEW were receiving oral nutritional supplements (ONS) and 8% intradialysis parenteral nutrition (IDPN), while 22% of the patients not catalogued with PEW were receiving ONS and 1% IDPN”.

Although the lack of nutritionists is well known (1), facilitating the diagnosis may encourage different professionals to set an earlier intervention. A future study of our group could be based in different approaches (intensive nutritional counselling, oral supplementation, or nutrition support by enteral or parenteral routes) as early intervention based on the results of the Nutrendial web application and its association with hard outcomes as mortality and morbidity in dialysis (2).

  1. Wang AY, Okpechi IG, Ye F, Kovesdy CP, et al. Assessing Global Kidney Nutrition Care. Clin J Am Soc Nephrol. 2022 Jan;17(1):38-52. doi: 10.2215/CJN.07800621. Epub 2022 Jan 3. PMID: 34980675; PMCID: PMC8763143.
  2. Burrowes JD, Larive B, Chertow GM, et al. Hemodialysis (HEMO) Study Group. Self-reported appetite, hospitalization and death in haemodialysis patients: findings from the Hemodialysis (HEMO) Study. Nephrol Dial Transplant. 2005 Dec;20(12):2765-74. doi: 10.1093/ndt/gfi132. Epub 2005 Oct 4. PMID: 16204298.

Related with Figure 2, a sentence has been added to clarify that “Figures shown in results box are examples only”.

  1. In ROC curve analysis for MIS score and other nutritional parameters such as albumin, BMI…etc, but these traditional bio-serological makers have already known less sensitivity. Could the authors provide the ROC curse for new web diagnostic tool?

Response 5: This computer tool was not designed as a new score, but uses already known scores. It computerizes them to make PEW diagnosis much easier, reinforcing the concept that MIS -also calculated with this web application- is the most accurate score to predict PEW (3)

  1. Borges MC, Vogt BP, Martin LC, Caramori JC. Malnutrition Inflammation Score cut-off predicting mortality in maintenance hemodialysis patients. Clin Nutr ESPEN. 2017 Feb;17:63-67. doi: 10.1016/j.clnesp.2016.10.006. Epub 2016 Nov 23. PMID: 28361749.

  1. Discussion section should focus on this new web tool and what is the advantage or disadvantage compared to traditional tools?

Response 6: As the reviewer suggests, a paragraph describing the advantages in comparison with other tools has been added to the Discussion section, as follows:

“In comparison with other tools, Nutrendial is easy to use, decreases the margin of human error, -as the calculations of PEW, MIS and SGA variables are automatic-, and consequently the time employed to calculate the nutritional status of CKD patients. Individual measurements remained for every patient, to follow their evolution and generate a final report that can be given to the patient or other professionals. The exploitation of data is allowed immediately and by centre, facilitating multicentre studies”.

Minor point:

  1. In line 136, there is a “space” between catalogued and with. Please correct it.

 Response 7: As the reviewer suggests, the space has been corrected.

  1. This web tool is not open to assess and needs to be registered. Is it only a registration system in Catalonia Society of Nephrology? Can it apply to other countries? 

Response 8: Nutrendial web application was initially offered at the regional level; currently it has already been offered at the national level and we were expecting the publication of this study to offer it to the international community. A paragraph clarifying this aspect has been added to the Discussion section, as follows:

“It is currently accessible only to Spanish health professionals, but will be translated to English in order to offer it to the international community”

Author Response

Response to the comments of reviewer 2.

Thank you very much for your comments and suggestions which we have taken into consideration and we have now incorporated into the revised manuscript.

Major comments

  1. The study was based on the development of a new tool, “Nutrendial”. However, it seemed difficult to use for an English reader, visiting their web link.

Response 1.1: To predict PEW, Nutrendial application web applies an algorithm where the variables needed to calculate PEW according to the classical criteria (12) for each body nutrition compartment have been previously introduced, specifically:

  • For the biochemical parameters section, albumin < 3.8 g/dL or cholesterol <100 mg/dL were used
  • For the body mass section, BMI <23 kg/m2 or a weight loss >10% in the last 6 months were used
  • For the muscle mass section, creatinine < 8mg/dL was used (13)
  • For the dietary intake section, nPCR <0.8 g/kg/day was used

This information has been included in the main text, in Methods and references have been updated.

In my view, a detailed description is needed in the method part on how to develop the tool and how it works so that it could be implemented in other places and countries as well.

Response 1.2: As the reviewer suggest, a paragraph describing the advantages in comparison with other tools has been added to the Discussion section, as follows:

“In comparison with other tools, Nutrendial is easy to use, decreases the margin of human error, as the calculations of PEW, MIS and SGA variables are automatic, and consequently the time employed to obtain the nutritional status of CKD patients. Individual measurements remained for every patient, to follow the evolution and generates a final report that can be given to the patient or other professionals. The exploitation of data is allowed immediately and by center, facilitating multicenter studies”.

Regarding the access for an English reader, Nutrendial web application was initially offered at a regional level, currently it has already been offered at a national level and we were expecting the publication of this study to offer it to the international community. A paragraph clarifying this aspect has been added to the Discussion section, as follows:

“It is currently accessible only to Spanish health professionals, but will be translated to English in order to offer it to the international community”.

  1. Subjective Global Assessment (SGA) is a part of the Malnutrition Inflammation Score (MIS) and to assess the nutritional status of dialysis patients, ISRNM proposed PEW criteria (at least 3 out of 4 criteria) [1] along with MIS will be sufficient to initially diagnose PEW patients [2].

Response 2.1: As the reviewer suggests, PEW criteria proposed by ISNRM along with MIS would probably be enough to assess the nutritional status of the dialysis patients, but SGA traditionally has been one of the most used tools, and since Nutrendial- when calculating MIS- collects the necessary variables to calculate SGA, we found it useful to offer this score for those who see it useful to continue using it.

A detailed description of how they collected anthropometric and dietary data, if any training session was organized should be added to improve the understanding of the readers.

Response 2.2: Anthropometric data collected were: dry weight, weight loss in 6 months and height, as well as the BMI derived from these.

Regarding to the description of how anthropometric and dietary data should be collected, the form in Figure 1 detailed how to introduce them in the Nutrendial web application, step by step (no difference to written SGA or MIS scoring system).

This sentence has been added to the main text in Methods: “In the physical examination section, visual support (photographs) is available in each answer option to aid in decision-making”.

Regarding to the training sessions, individual support was offered to those who requested it, speeches have been given at a local level, and also at regional and national congresses.

Minor comments

  1. Overall, this is a well-explained study with few writing errors/typos and I would suggest hiring a professional proof editor.

Response 1: As the reviewer suggests, the main text has been fully reviewed by a native-speaker colleague and the writing errors were corrected.

  1. For instance, in the abstract, in line 26, PD should be written instead of DP. In the method section, in line 87, it should be renal replacement therapy in line 97, “facilitate” instead of “facility”. In the discussion part, in line 246, IMC>30 kg/m2 is mentioned. It should be corrected to BMI. In all the tables, commas were used for numerical values, such that, serum albumin value of 3.83 g/dl instead of 3,83 g/dl.

Response 2: Typewriting errors have been corrected in the text

  1. The authors also mentioned in their method part that they had collected dietary information from patients. This is an important diagnostic criterion for assessing PEW (ISRNM) [1]. However, no explanation on how they had collected dietary data (which method was used-food frequency questionnaire (FFQ) [3] or 24-hour diet recall day) [4], and thus no quantitative data were provided to show the dietary intake of patients on daily basis and only qualitative diet-related information was demonstrated (type of food intake).

Response 3: We appreciate the comment from the reviewer. There is a specific question about dietary intake in the web application in a qualitative way required for MIS and SGA calculation, but not a specific questionnaire. Together with our nutritionist we are evaluating to include, in a short-term, a more complete section on dietary intake, where data on energy and protein intake could be included. These data would have been previously calculated in an external nutritional calculation program. The calculation would be based on a 24-hour diet recall day or a 3-day food diary.

  1. In the result section, the cut-off for MIS was 6 (line 134), however, in table 1, MIS<8 was used (no reference mentioned) MIS [5].

Response 4.1: In the results section, MIS 6 is the median, not the cut-off point that was calculated afterwards. In the table, we decided to set MIS <8, in line with relevant articles which have been published previously:

Rambod M, Bross R, Zitterkoph J, Benner D, Pithia J, Colman S, Kovesdy CP, Kopple JD, Kalantar-Zadeh K. Association of Malnutrition-Inflammation Score with quality of life and mortality in hemodialysis patients: a 5-year prospective cohort study. Am J Kidney Dis. 2009 Feb;53(2):298-309. doi: 10.1053/j.ajkd.2008.09.018. Epub 2008 Dec 13. PMID: 19070949; PMCID: PMC5500250.

 In line 164 and in line 165, it was written that, patients with PEW received higher nutritional intervention, but no explanation on how and when it was done.

Response 4.2: As the reviewer suggests, we have now clarified how the nutritional interventions were done related to patients with and without PEW. These data were provided by the dialysis centers when the study was performed and the question about when or when was initiated, was not included. The authors plan to perform an analysis of these data to know in a deeper way the profile of nutritional intervention in professionals who have continued to use the Nutrendial tool.

The following sentence has been included in the results section: More specifically, 53% patients with PEW were receiving oral nutritional supplements (ONS) and 8% intradialysis parenteral nutrition (IDPN), while 22% of the patients not catalogued with PEW were receiving ONS and 1% IDPN”.

  1. Other comments: As the authors mentioned different nutritional scores in the title, Kidney Disease Quality of Life (KD-QoL), SF-36 score [6], appetite and diet analysis tool (ADAT) score[7], and restless leg syndrome (RLS) score [8] could also be added to that newly developed software to better understand the overall physiological and psychological status of those patients which in turn reflect on their nutritional well-being. Overall, early diagnosis and management of PEW among dialysis patients is necessary for their better survival and in this context, the current study took a holistic approach to diagnose 23% of PEW patients in its community in a simple and feasible way which helped them to get nutritional intervention on time.

Response 5: As the reviewer suggests, it would be very interesting to add in the short term in the Nutrendial web application the ADAT score to evaluate appetite as part of the routine nutritional assessment in dialysis patients, knowing that the HEMO study demonstrated that appetite assessment can be useful prognostically. In the long term, the authors would also include psychological and functional assessment tools.

Round 2

Reviewer 1 Report

The authors have responded to my comments well, and I have no further comments about this article.